# Transcriptomic profiling of pancreatic neuroendocrine tumors: dysregulation of WNT, MAPK, PI3K, neddylation pathways and potential non-invasive biomarkers

**Helvijs Niedra**[1,☉], **Olesja Rogoza**[1,☉], **Rihards Saksis**[1], **Raitis Peculis**[1], **Anzela Halilova**[1], **Aija Gerina**[2], **Sofija Vilisova**[2], **Natalja Senterjakova**[2], **Aldis Pukitis**[2], **Ignacio Ruz-Caracuel**[3], **Julie Earl**[4], **Georgina Kolnikova**[5], **Peter Dubovan**[6,7], **Miroslav Tomas**[6,7], **Peter Makovicky**[7], **Maria Urbanova**[7], **Bozena Smolkova**[7], **Eythimios Koniaris**[8], **Ioanna Aggelioudaki**[9], **Agapi Kataki**[10], **Vita Rovite**[1]*

1 Latvian Biomedical Research and Study Centre, Ratsupites Str, Riga, Latvia, 2 Pauls Stradins Clinical University Hospital, Pilsonu Str, Riga, Latvia, 3 Department of Pathology, Ramón y Cajal University Hospital, Ctra. Colmenar Viejo, Madrid, Spain, 4 Ramón y Cajal Health Research Institute (IRYCIS), Ramón y Cajal University Hospital, Ctra. Colmenar Viejo, Madrid, CIBERONC, Spain, 5 Department of Pathology, National Cancer Institute, Klenová, Bratislava, Slovak Republic, 6 Department of Surgical Oncology of Slovak Medical University in Bratislava, National Cancer Institute, Klenová, Bratislava, Slovak Republic, 7 Biomedical Research Center, Slovak Academy of Sciences, Dúbravská cesta, Bratislava, Slovak Republic, 8 Department of Pathology, Hippokration General Hospital of Athens, Vasilissis Sofias, Athens, Greece, 9 Second Department of Surgery, Aretaieio University Hospital, Vasilissis Sofias, Athens, Greece, 10 First Department of Propaedeutic Surgery, Hippokration General Hospital of Athens, Vasilissis Sofias, Athens, Greece

☉ These authors contributed equally to this study
* vita.rovite@biomed.lu.lv

## Abstract

The study aimed to identify altered signaling pathways and potential non-invasive biomarkers for pancreatic neuroendocrine tumors (PanNETs) through transcriptomic profiling of tumor tissues. The analysis encompassed samples from non-functional PanNETs (NF-PanNETs), insulinomas, and tumor-adjacent pancreatic tissues (TAPT). In the differential expression analysis comparing PanNETs and TAPTs, we identified 1,210 differentially expressed genes at a false discovery rate significance threshold of < 0.05 and with Log2FoldChange values of > 0.5 and <−0.5. Further pathway enrichment analysis revealed a multitude of overrepresented signaling pathways related to cell proliferation, survival, and tumorigenesis. Significant findings included the Beta-catenin-independent and TCF-dependent WNT signaling pathways, MAPK1/MAPK3 signaling, and terms associated with PI3K/AKT/mTOR signaling. Among the list of DEGs, we also identified 28 upregulated genes encoding cell surface proteins and 24 upregulated genes encoding cancer-associated secretome proteins. Since the proteins of these genes are found in the bloodstream, there is potential for further testing of these markers as biomarkers for liquid biopsy assays. Overall, these findings underscore the promise of transcriptomic landscape analysis

**Data availability statement:** Salmon transcript count data and raw sequencing data for each sample included in this study are available at the NCBI Gene Expression Omnibus (GEO) repository under the accession no.: GSE281039. Any additional datasets and source code used to generate figures displayed within the paper are listed in Supplementary material.

**Funding:** This research was supported by the European Regional Development Fund project "Establishing an algorithm for the early diagnosis and follow-up of patients with pancreatic neuroendocrine tumors – NExT" project (1.1.1.5/ERANET/20/03) and the EUs Horizon 2020 research and innovation program (grant No. 857381/VISION). Partners from Slovakia were supported by the Slovak Academy of Sciences, Slovakia (NExT-0711) and the Slovak Research and Development Agency (APVV-21-0197, APVV-20-0143). Partners from Greece received funding from the TRANSCAN-2 program ERA-NET JTC 2017 "Translational research on rare cancers" within the project NExT (NKUA): Ministry of Education, Research and Religious Affairs - General Secretariat for Research and Technology, Greece (T9EPA3-00012). Lastly, during the manuscript preparation and review process, Rihards Saksis was supported by the project "Strengthening of the capacity of doctoral studies at the University of Latvia within the framework of the new doctoral model", identification No.8.2.2.0/20/I/006. During the study Helvijs Niedra was supported within the framework of the European Union's Recovery and Resilience Mechanism project No.5.2.1.1.i.0/2/24/I/CFLA/001 "Consolidation of the Latvian Institute of Organic Synthesis and the Latvian Biomedical Research and Study Centre" (Grant subsection identifier: ANM_K_DG_07).

**Competing interests:** The authors declare no conflict of interest.

in identifying PanNET-specific non-invasive biomarkers and uncovering potential therapeutic targets.

## Introduction

Pancreatic neuroendocrine tumors (PanNETs) are a heterogeneous group of neoplasms that arise from the neuroendocrine cells of the pancreas, demonstrating various clinical manifestations that contribute to a significantly increased risk of mortality [1,2]. Hormonally, PanNETs are divided into hormonally active and non-functional (NF-PanNETs), with the majority of tumors (at least 70%) belonging to the NF-PanNET category [3,4]. Histologically, PanNETs are classified as well-differentiated tumors. Based on the Ki-67 and mitotic rate, PanNETs can be subdivided into three tiers - grade 1 (G1), grade 2 (G2), and grade 3 (G3), with G3 showing the highest proliferative capacity [5]. Epidemiologically PanNETs are rare, with an approximate annual incidence of 1 per 100'000 [6]; yet they present a major healthcare challenge as in up to 21–65% of cases the disease presents with metastatic involvement [6]. The median overall survival for patients with PanNETs is 68 months [7]. This is further aggravated by the lack of treatment options for patients with unresectable PanNETs [2].

Similar to other cancers, surgical resection continues to offer the most favorable clinical outcomes for patients with PanNETs [3,8,9]. For unresectable and recurring cases the first-line treatment involves the administration of somatostatin analogs (SSAs) as well as targeted treatment agents (Everolimus and Sunitinib) [10,11]. This raises the need for early detection of recurrence after curative resection (occurring in 20% of patients) and expanded medical treatment options for unresectable and recurrent cases [12].

The currently used serum biomarkers (chromogranin A, neuron-specific enolase, and pancreatic peptide) are non-specific for PanNETs with limited sensitivity and specificity. As such, the diagnosis, prognosis, and optimal clinical management strategy for these tumors remain challenging [2]. NF-PanNETs typically grow asymptomatically because the amount of produced hormones do not cause clinically relevant symptoms. As a result, the majority of NF-PanNETs are frequently misdiagnosed or diagnosed in advanced stages [9].

The molecular pathogenesis and alterations of sporadic PanNETs are still only partially understood, despite significant achievements in the past decade. Sequencing studies have shown that PanNETs usually contain mutations in genes related to mammalian target of rapamycin (mTOR) signaling, including *PTEN*, *TSC2*, *PIK3CA*, and *MEN1*, which is associated with multiple neuroendocrine neoplasia type 1 [13–15]. Other proposed genes linked to PanNETs development are *ATRX* and *DAXX*, which are mutated in up to 60% of PanNETs [14,16,17]. Histopathological studies have shown that a loss of nuclear ATRX/DAXX expression promotes alternative lengthening of telomeres and is associated with significantly reduced 5-year relapse-free survival (40% vs. 80%) [18,19]. Gene expression-based studies have shown that gastroenteropancreatic NETs, in general, overexpress somatostatin

receptors [20] as well as genes related to vascular endothelial growth factor (VEGF) signaling [21] and mTOR signaling for which various treatments have been already developed [22,23].

Further exploration of the PanNET transcriptome is essential to facilitate the discovery of novel specific biomarkers and therapeutic targets that can enhance stratification, guide prognosis, and treatment decisions. Consequently, in this study, we set out to perform whole-transcriptome RNA sequencing to identify differentially expressed genes (DEGs) between PanNETs and tumor-adjacent pancreatic tissues (TAPTs), as well as between different tumor grades, which was followed by comprehensive pathway enrichment analysis and the search for potential cell surface markers (CSMs) and secretome proteins (SPs). Overall, this research provides insights into the molecular mechanisms underlying the development of PanNETs and outlines potential targets for blood-based assay development.

## Materials and methods

### Study group and design

The study, written informed consent template, and patient inclusion were approved by the Central Medical Ethics Committee of Latvia protocol (No. 1.1–2/67); Ethics Committee of National Cancer Institute, Slovakia, protocol (No. 2019-02-20-100259); Ramon y Cajal Ethical and Scientific Committees, protocol (No. 196–19); Scientific Committee General Hospital of Athens, (No. 34/14-7-2020); Aretaieio Hospital Athens Medical School (NKUA) Research and Ethics Committee, (No. 10-7-2020). All patients provided a signed informed consent prior to the participation within the study. Following the ethical approval, the patient recruitment and FFPE sample collection was done throughout the period of January 2021 till October 2021. The patient recruitment was done via the following institutions: Pauls Stradins Clinical University Hospital (Riga Latvia), National Cancer Institute (Bratislava, Slovakia), Ramón y Cajal University Hospital (Madrid, Spain), and Hippokration General Hospital of Athens (Athens, Greece). The compiled clinical data of the PanNET patients are presented in S1 Table. In total, 36 FFPE samples (30 PanNET, 6 TAPT) were sequenced from 30 patients (Fig 1). Clinical data collection included general information (sex, age at surgery, tumor grade, hormonal type, Ki-67 index, treatment before surgery) and did not include any information that could identify individual participants during or after data collection. The cohort

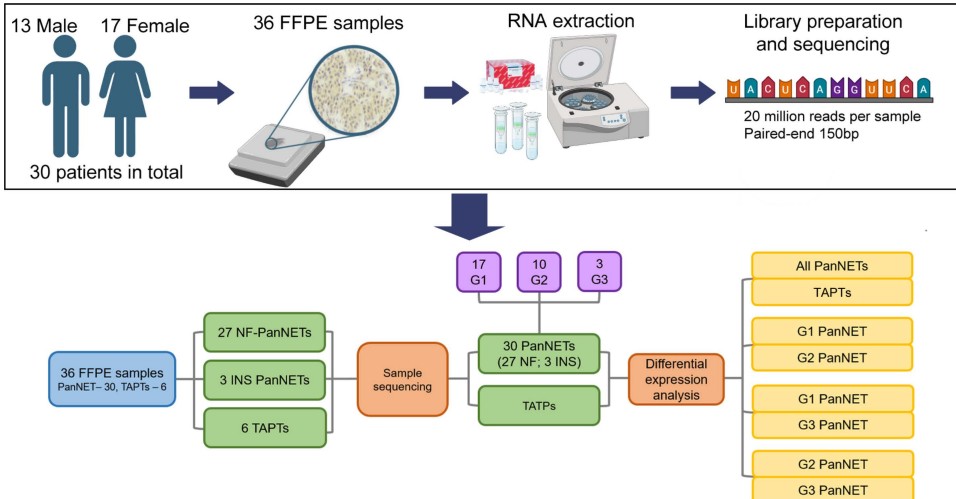

**Fig 1. Schematic overview of study design encompassing initial sample size, sample groups, and differential expression analysis comparisons.** Sequencing data quality control included batch effect assessment using principal component analyses. Abbreviations: FFPE – formalin fixed paraffin embedded, RNA – ribonucleic acid, PanNET – pancreatic neuroendocrine tumor, NF-PanNET – non-functioning pancreatic neuroendocrine tumor, TAPT – tumor adjacent pancreatic tissue, INS – Insulinoma, G1 – grade 1, G2 – grade 2, G3 – grade 3.

consisted of 17 female patients and 13 male patients and represented all three tumor grades. A total of five differential gene expression (DGE) analyses were performed (Fig 1). Due to the low sample size of INS tumors (one G1, two G2 cases), we decided to combine NF-PanNETs with INS as the primary focus of our study was to assess the changes in transcriptomic profiles of PanNETs in comparison to non-tumor tissues and evaluate the differences between three grades of PanNETs.

## Transcriptome sequencing

RNA for RNA-seq was extracted from the FFPE samples using the RNeasy FFPE Kit (Qiagen, Germany) following the manufacturer's instructions. The concentration of extracted RNA was measured with the Qubit RNA HS Kit (Thermo Fisher, USA) using the Qubit 2.0 Fluorometer (Thermo Fisher, USA). Extracted RNA was evaluated using Agilent 2100 Bioanalyzer with the RNA 6000 Pico Kit (Agilent Technologies, USA). For library preparation cDNA synthesis and cDNA amplification was carried out using the REPLI-g WTA Single Cell Kit (Qiagen, Germany) with the Poly A+ mRNA amplification protocol. cDNA was then enzymatically fragmented and RNA-seq libraries were prepared using the MGIEasy PCR–Free DNA Library Prep Set (MGI, PRC). Paired-end sequencing was carried out on the DNBSEQ-G400 platform (MGI, PRC) with 20 million reads per sample and 150 bp read length.

## Data analysis

Quality control of raw sequencing data was performed using FastQC (v0.11.9) and MultiQC (v1.12) software. The paired-end reads were trimmed with fastp software (v0.23.2) reads with quality score below 20 and read length below 75 base pairs were discarded. The read count and quality of trimmed reads were again inspected by the above-mentioned QC software. The ribosomal RNA (rRNA) was removed using SortMeRNA (v2.1), followed by final quality and read count evaluation. Reads were then quasi-mapped and quantified with Salmon software (v1.9.0) against GENCODE (v38) Homo sapiens genome. R (v4.2.2) was used to summarize gene-level counts with the tximeta package (v1.16.0).

DGE analysis was performed using DESeq2 (v1.38.3) [24]. Counts were filtered by frequency, setting the count threshold at 10 and the minimum sample frequency threshold at 3. The Wald test was used to determine gene expression differences. The default filtering function was replaced by independent hypothesis weighting from the IHW package (v1.26.0). The $Log_2$FoldChange ($Log_2$FC) shrinkage algorithm (from the apeglm package (v1.20.0) [25]) was used to adjust $Log_2$FC values for genes with low counts and high dispersions. DEGs heatmaps were plotted using pheatmap package (v1.0.12). Volcano plots were plotted using EnhancedVolcano package (v1.16.0).

Pathway enrichment analyses were performed using the PathfindR package (v2.4.1) – an active subnetwork-oriented enrichment analysis (R package v2.4.1) developed by Ulgen et al. [26]. The processes of protein-protein interaction (PPI) network generation and active subnetwork search in PathfindR were performed using the STRING database (v12.0) [27]. Pathway enrichment analysis with the identified subnetworks was then performed using the Reactome database (v89). The Reactome database was chosen because, according to a methodological study by Gable et al., it provided the most optimal overall performance among manually curated annotation systems [28]. Genes in the generated PPI network were used as background genes for the overrepresentation analysis. The analysis was done over 10 iterations with "greedy algorithm" as search algorithm. Pathways with false discovery rate (FDR) adjusted p-values of <0.05 were considered as significant. Lastly, to identify groups of biologically similar pathways a hierarchical clustering analysis was performed using PathfindR package.

To determine whether DEGs represented CSMs or SPs, DEGs from tumor vs. TAPT comparison were compared to the list of genes in silico human surfaceome [29] and human protein atlas (Secreted proteins predicted by MDSEC list) [30]. Identified SPs were tested for significant tissue enrichment using the TissueEnrich (v1.18.0) package, available in the Bioconductor project [31]. To examine candidate CSMs and SPs in other PanNET datasets we used publicly available data from Gene Expression Omnibus. The accession numbers for included studies were GSE98894 and GSE116356. Data

from study GSE98894 contained gene expression data from 83 fresh-frozen PanNET (primary tumor) samples and 29 liver metastasis samples. Study GSE116356 contained gene expression from eight FFPE PanNET primary tumor tissues. The expression data from both studies was combined with our data and imported in DESeq2 for variance stabilizing transformation. Following this a batch effect correction was performed using limma package (v3.60.2) to adjust for batch effect introduced by difference in studies.

A detailed overview of R code for DESeq2, pathway enrichment, and tissue enrichment analysis is provided in S1 Text. Salmon transcript count data and raw sequencing data for each sample included in this study are available at the NCBI Gene Expression Omnibus (GEO) repository under the accession no.: GSE281039.

## Results

### Transcriptomic changes in pancreatic neuroendocrine tumor tissues

To identify the specific genes and pathways that are dysregulated in PanNET tissues and potentially related to tumorigenesis, we performed a DGE analysis by comparing all PanNET samples (27 NF-PanNETs, 3 INS; 17 G1, 10 G2, and 3 G3 tumors) against six TAPT samples, which included both exocrine and endocrine parts of the pancreas. In the principal component analysis plot it can be observed that there are differences in gene expression between TAPTs and PanNETs based on top 500 genes with the highest variance (Fig 2A). It should be noted that two samples (NET22 and NET23) exhibit gene expression patterns close to TAPTs indicating a possible contamination of non-tumor tissue. According to distribution of P values (Fig 2B) and changes shown by volcano plot (Fig 2C) the DGE analysis did identify significant differences between tumor and TAPT samples. At FDR < 0.05 and $Log_2FC > 0.5$ thresholds, significant dysregulation was observed for 1210 genes (S2 Table). Of the DEGs, 86% (n = 1041) were upregulated, while 169 (14%) were downregulated. The median $Log_2FC$ value for the upregulated genes was 2.9 (interquartile range [IQR] = 1.7), whereas for the downregulated genes, it was −2.5 (IQR = 1.5). A clustering analysis based on top 25 upregulated and top 25 downregulated genes, according to $Log_2FC$ values, shows a clear segregation between tumor samples and TAPT samples. However, the expression of these genes could not distinguish between tumor grades (Fig 2D).

### Overrepresented pathways in the list of genes differentially regulated within PanNETs

By subjecting the entire list of 1,210 DEGs to the active subnetwork-oriented enrichment analysis using the STRING database (v12.0), 289 genes were filtered out because they represented non-coding RNAs and transcribed pseudogenes (60 entries), while 229 genes had no entries in the STRING database. Using the remaining 921 genes, the pathfindR analysis identified 14 active PPI subnetworks. Downstream Reactome pathway enrichment analyses with each of the 14 PPI subnetworks resulted in a total of 199 statistically significant (FDR < 0.05) overrepresented Reactome pathways (S3 Table).

By applying the clustering analysis to identify biologically similar pathways (defined by the number of overlapping genes), a set of 40 clusters was identified. Of the 40 clusters, the top five largest clusters (by the number of pathways) were cluster 1 (78 terms), cluster 19 (14 terms), cluster 16 (13 terms), cluster 8 (10 terms), and cluster 11 (8 terms) (Fig 3A). To simplify the visualization and interpretation of the results, we chose to further focus on the top five pathways (according to FDR value) within each of the largest clusters (1, 19, 16, 8, and 11)—a total of 25 pathways (Fig 3B). From this perspective, it can be observed that the top five pathways in each of the five clusters are mostly upregulated in tumors compared to TAPT (Fig 3C and Fig 3D). As illustrated by the PPI network (Fig 3E), the genes from these 25 pathways are biologically interconnected, with 186 physical interactions between 86 proteins.

The majority of terms under clusters 1, 16, and 8 were related to oncogenic signaling cascades. In the largest cluster—cluster 1—neddylation (R-HSA-8951664) was identified as the representative term consisting of 21 upregulated DEGs (Fig 3D). Neddylation has been shown to be upregulated in a multitude of cancers [32,33]; with studies showing that it can directly impact the activity of NET related tumor suppressor proteins, e.g., PTEN [34] and MEN1 [35]. To elucidate

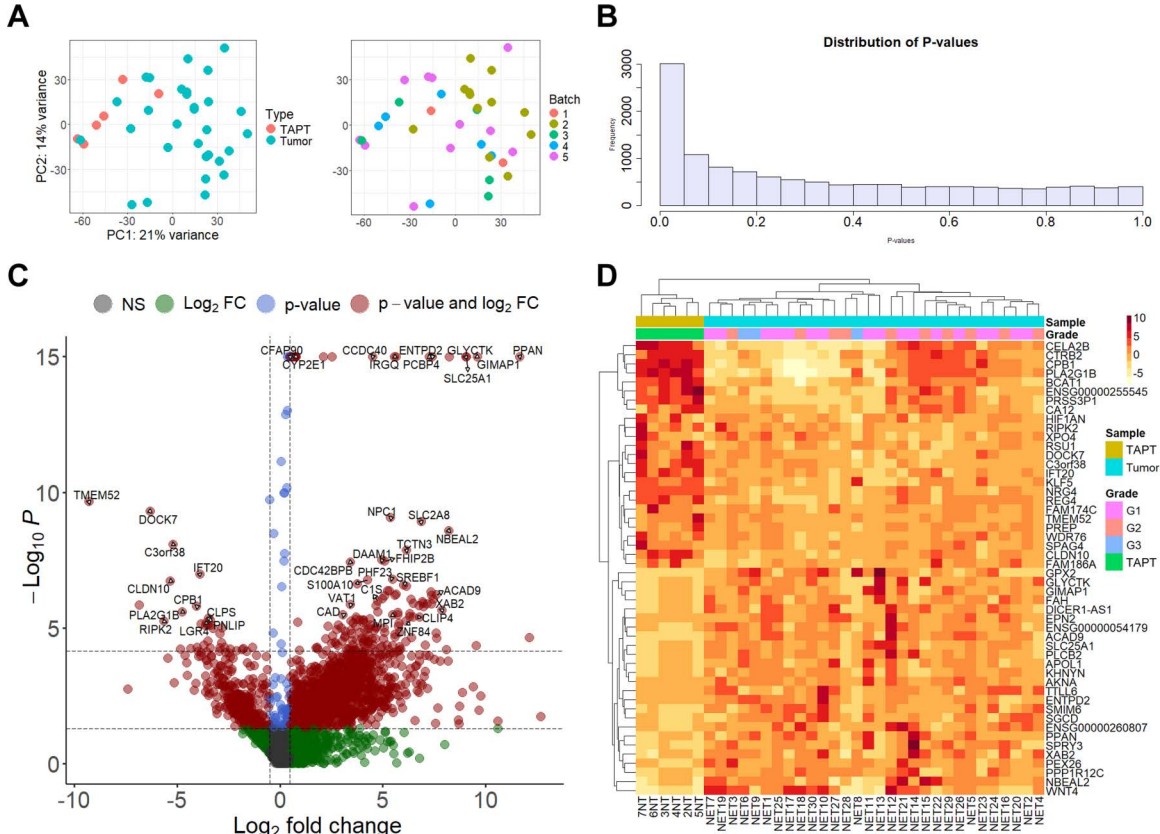

**Fig 2. Results of differential gene expression (DGE) analysis between pancreatic neuroendocrine tumor and tumor-adjacent pancreatic tissues.** A – scatterplot of principal component analysis results for tumor (n = 30) and TAPT tissues (n = 6). Analysis was based on the top 500 genes with the highest variance. B – histogram showing distribution of p-values from results of differential expression analysis using DEseq2; the plot shows shows major differences between tumor and TAPT samples; X-axis – p-value; Y-axis – frequency. C – Volcano plot; Y-axis – -Log$_{10}$p-values; X-axis – Log$_2$Fold-Change (Log$_2$FC) values from the DESeq2 analysis. Y-axis is capped at -Log$_{10}$P = 10 for visual clarity. Horizontal dashed lines represent the p-value (0.05), and false discovery rate (FDR) adjusted p-value (0.05) thresholds. Vertical dashed lines represent Log$_2$FC thresholds (+/- 0.5). NS – genes below both thresholds. D – heatmap visualization of mean normalized expression values for the top 25 upregulated and downregulated differentially expressed genes according to Log$_2$FC values. Clustering distance measure – Euclidean.

additional context of all 21 upregulated DEGs involved in neddylation we performed additional STRING functional protein-protein interaction analysis. Here, according to k-means clustering, two clusters can be observed (S1 Fig). By examining terms related to Gene Ontology (GO) cellular components and biological processes DEGs from cluster 1 are primarily associated with proteasome complex (GO:0000502), ubiquitin ligase complex (GO:0000151), and proteolysis involved in protein catabolic process (GO:0051603). On the other hand, DEGs from cluster 2 primarily represent cullin-RING ubiquitin ligase complex (GO:0031461) and protein ubiquitination process (GO:0016567). These results are intriguing as they indicate that proteasomal degradation seems to be upregulated in PanNETs.

The other most notable terms here in cluster 1 (according to FDR value), aside from neddylation, were pathways related to WNT signaling, ubiquitination, and MAPK1/MAPK3 signaling (Fig 3B). In our previous spatially resolved transcriptome profiling study, where eight PanNETs were compared against islet cells, Wnt signaling pathways were also found to be overrepresented in G2/G3 tumors. The overrepresented Reactome pathways were: TCF-dependent signaling in response to WNT and Beta-catenin-independent WNT signaling, which are also overrepresented in this study (S2 Fig) [36]. In a different study by Jiang et al. it was proposed that Wnt signaling is altered in MEN1 (menin)-defective tumors,

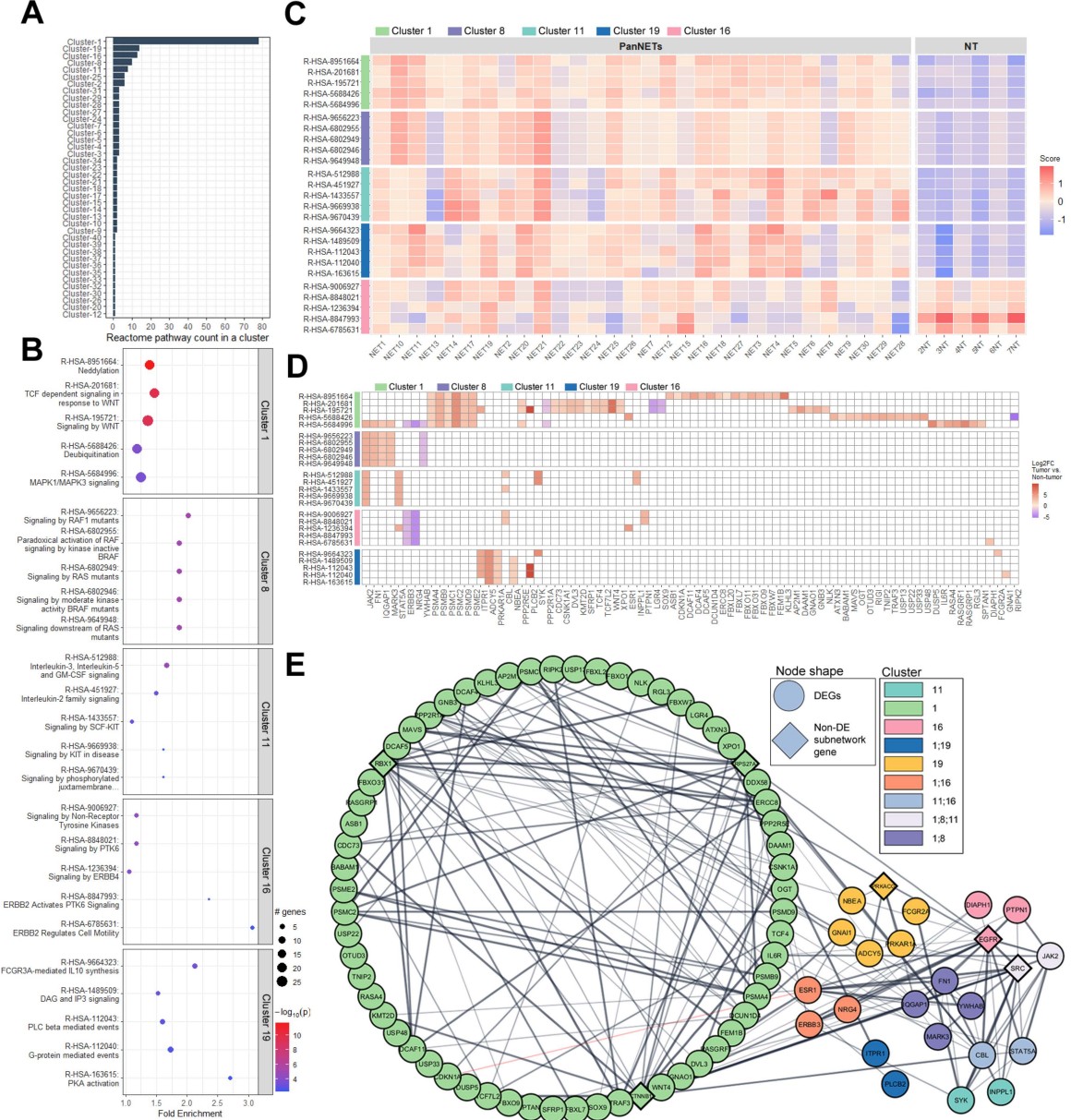

**Fig 3. Results of active subnetwork-oriented enrichment analysis for the differentially expressed genes (DEGs) identified in tumor vs. tumor-adjacent pancreatic tissue (TAPT) analysis.** Figures include the top five pathways (based on false discovery rate p-value) from each of the five largest clusters. A – bar chart of Reactome pathways per cluster; X-axis – pathway count in a cluster. B – enrichment chart representing enrichment results; X-axis – fold enrichment value; Y-axis – Reactome term identifiers followed by description. C – Heatmap of the agglomerated z-scores of enriched terms per sample. The plot displays how the selected 25 terms are altered in each sample based on variance stabilizing transformed (vst) normalized count data from 30 tumor and 6 TAPT samples. D – Terms by genes heatmap. The heatmap shows the upregulation or downregulation (Log$_2$FC) of genes (86 in total) related to a specific term. E – STRING physical protein-protein interaction network of 86 genes related to pathways displayed in figures B, C, and D. Network p-value – 1.0 x 10^16, based on 186 edges and 79 expected edges for a random set of proteins of the same size.

as menin is involved in the regulation of β-catenin activity via phosphorylation [37]. In our data (S3 Table) we can observe an overrepresentation of four β-catenin related pathways. Regarding the MAPK1/3 signaling we can also observe that in cluster 8 the representative term was Signaling by RAF1 mutants. The other RAS/RAF related terms in this cluster were

Signaling by RAS mutants and Signaling downstream of RAS mutants (Fig 3B and S3 Table). This shows that the entire RAS/RAF/MAPK signal transduction cascade is altered in PanNETs.

In cluster 16, the representative term was "Signaling by Non-Receptor Tyrosine Kinases" (R-HSA-9006927). Under this category, there were a multitude of overrepresented oncogenic signaling pathways, with the most notable being related to PTK6 and ERBB signaling events (Fig 3B).

Clusters, 19 and 11, consisted of pathways involved in immune regulation (Fig 3B). A particularly interesting case is cluster 19, where the representative term was FCGR3A-mediated IL10 synthesis (R-HSA-9664323), which points to an anti-inflammatory response. Other pathways in the cluster related to the anti-inflammatory response included: anti-inflammatory response favoring Leishmania parasite infection (R-HSA-9662851) and Leishmania parasite growth and survival (R-HSA-9664433) (S3 Table). In cluster 11, the representative term was "Interleukin-3, Interleukin-5 and GM-CSF signaling," with the other top five terms related to SCF and KIT signaling (Fig 3B), pointing to the regulation of immune cell proliferation, migration, and differentiation.

Regarding other notable pathways listed in S3 Table, there were also several significant signaling pathways related to regulation of PTEN (R-HSA-8948751, R-HSA-6807070, R-HSA-8943724), PI3K (R-HSA-2219528, R-HSA-1963642) and AKT (R-HSA-1257604), all of which are components of mTOR signaling. PI3K/AKT/mTOR signaling pathway and its role in the development of PanNETs has been well established [38,39]; as in 15% of PanNETs, mutations have been found in at least one of the following genes: *PTEN*, *PIK3CA*, and *TSC2* [14]. By examining pathways PI3K/AKT signaling in cancer, PIP3 activates AKT signaling, and PTEN regulation (S4 Fig), we can observe that the 21 genes are upregulated and 4 genes are downregulated. Amongst these are notable tumor related genes *AKT3*, *CDKN1A*, *ESR1*, FOXO1, *TSC2*, *ERBB3*, and *NRG4* (S4 Fig).

## Transcriptomic changes across different grades of PanNETs

The DGE analysis between G1 and G2 PanNET tissues resulted in 15 DEGs (10 upregulated, 5 downregulated) (Figs 4A, 4D, and S4 Table). The median $Log_2FC$ value for upregulated genes was 2.82 (IQR = 1.4), while for downregulated genes, it was −2.81 (IQR = 0.8). The most upregulated and downregulated genes were Nitric Oxide Synthase Interacting Protein (*NOSIP*, $Log_2FC$ = 4.1, FDR = 0.02) and EF-Hand Domain Family Member D2 (*EFHD2*, $Log_2FC$ = −3.4, FDR = 0.01). The DGE analysis between G3 and G1 PanNET tissues resulted in 121 DEGs (48 upregulated, 73 downregulated), with a median $Log_2FC$ of 4.5 (IQR = 1.6) for upregulated genes and a median $Log_2FC$ of −3.8 (IQR = 4.4) for downregulated genes (Figs 4B, 4E, and S4 Table). Here, the most upregulated and downregulated genes were lipocalin 2 (*LCN2*, $Log_2FC$ = 7.9, FDR = 0.002) and required for meiotic nuclear division 1 (*RMND1*, $Log_2FC$ = −11.1, FDR = 0.0001). Lastly, the analysis between G3 and G2 tumors resulted in the fewest DEGs, as only glutathione peroxidase (*GPX2*, $Log_2FC$ = 7.63, FDR = 0.015) and neural cell adhesion molecule (*NCAM1*, $Log_2FC$ = −5.655, FDR = 0.002) were found to be differentially regulated (Figs 4C and 4F). *NCAM1* was also downregulated in G3 vs G1 tumor comparison. Interestingly, *NCAM1* is another marker of neuroendocrine differentiation in NETs [40]. Accordingly, these results indicate that G3 NETs, while still showing histological features of well-differentiated tumors, are less differentiated than G2 and G1 NETs in terms of *NCAM1* expression.

## Surfaceome and secretome markers in PanNETs

CSMs and SPs have significant clinical implications due to their potential to provide valuable information about disease states, prognosis, treatment response, and targeted therapy options in a minimally invasive manner. For these reasons, our study also investigated whether the list of DEGs included genes encoding CSMs and SPs. To do this, we examined the list of 1210 DEGs from the tumors vs. TAPT analysis in the in silico human surfaceome [29] and human protein atlas [30] databases (Fig 5A). As a result, we identified a total of 31 CSM genes (27 upregulated, 4 downregulated) in the list of DEGs (S5 Table). The expression values of upregulated genes are compiled in Fig 5B. Four genes (*CEACAM1*, *EDNRA*,

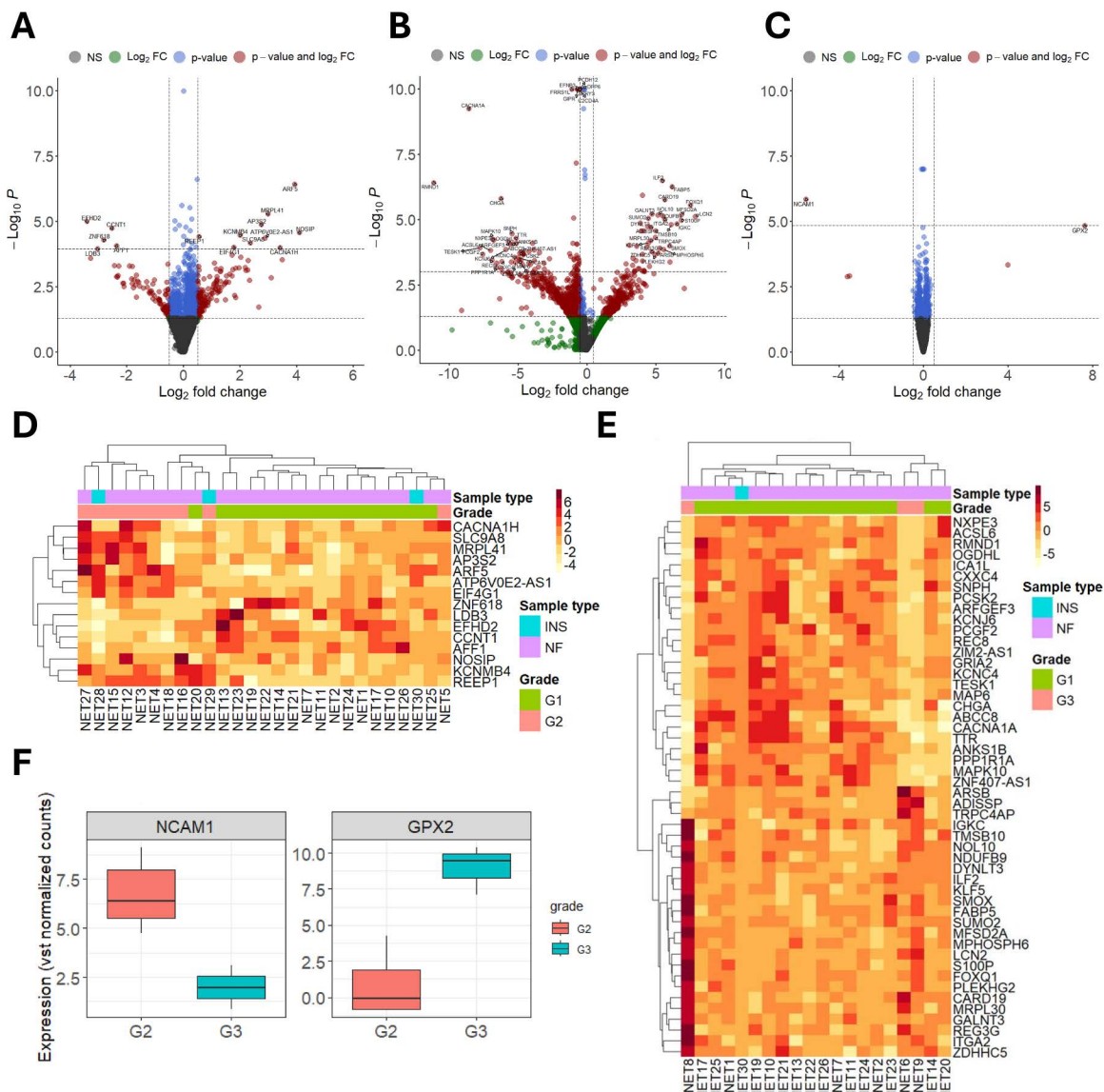

**Fig 4. Results of differential gene expression between different grades of pancreatic neuroendocrine tumors.** A, B, and C – Volcano plots; Y-axis – -Log$_{10}$p-values; X-axis – Log$_2$FoldChange (Log$_2$FC) values (x-axis) from the DESeq2 analysis. Y-axis is capped at -Log$_{10}$P = 10 for visual clarity. Horizontal dashed lines represent the p-value (0.05) and false discovery rate (FDR) adjusted p-value (0.05) thresholds. Vertical dashed lines represent Log$_2$FC thresholds (+/- 0.5). NS – genes below both thresholds. A – Grade 2 (G2) vs. Grade 1 (G1) tumors; B – Grade 3 (G3) vs. G1 tumors; C – G3 vs. G2 tumors. D and E – Heatmaps of mean normalized expression values for the top 25 upregulated and downregulated differentially expressed genes (FDR < 0.05) according to Log$_2$FC values. Sample type annotations: insulinomas (INS), non-functioning (NF). D – G2 vs. G1 tumors; E – G3 vs. G1 tumors. F – Boxplots representing the two differentially expressed genes (FDR < 0.05) in the G3 vs. G2 tumors comparison; Y-axis – variance-stabilizing transformed (VST) normalized count values.

*IFNAR1*, and *SLC8A1*) had at least one entry in the DrugBank database of approved drugs. Notable entries related to cancer treatment were: 1) technetium (99mTc) arcitumomab (DB00113), a technetium-labeled carcinoembryonic antigen (CEA) monoclonal antibody used in the treatment of CEA-overexpressing colorectal cancers; 2) natural human interferon alfa and recombinant human interferon alfa-2b (DB05258 and DB00105), used to treat viral infections and blood cancers, including melanoma.

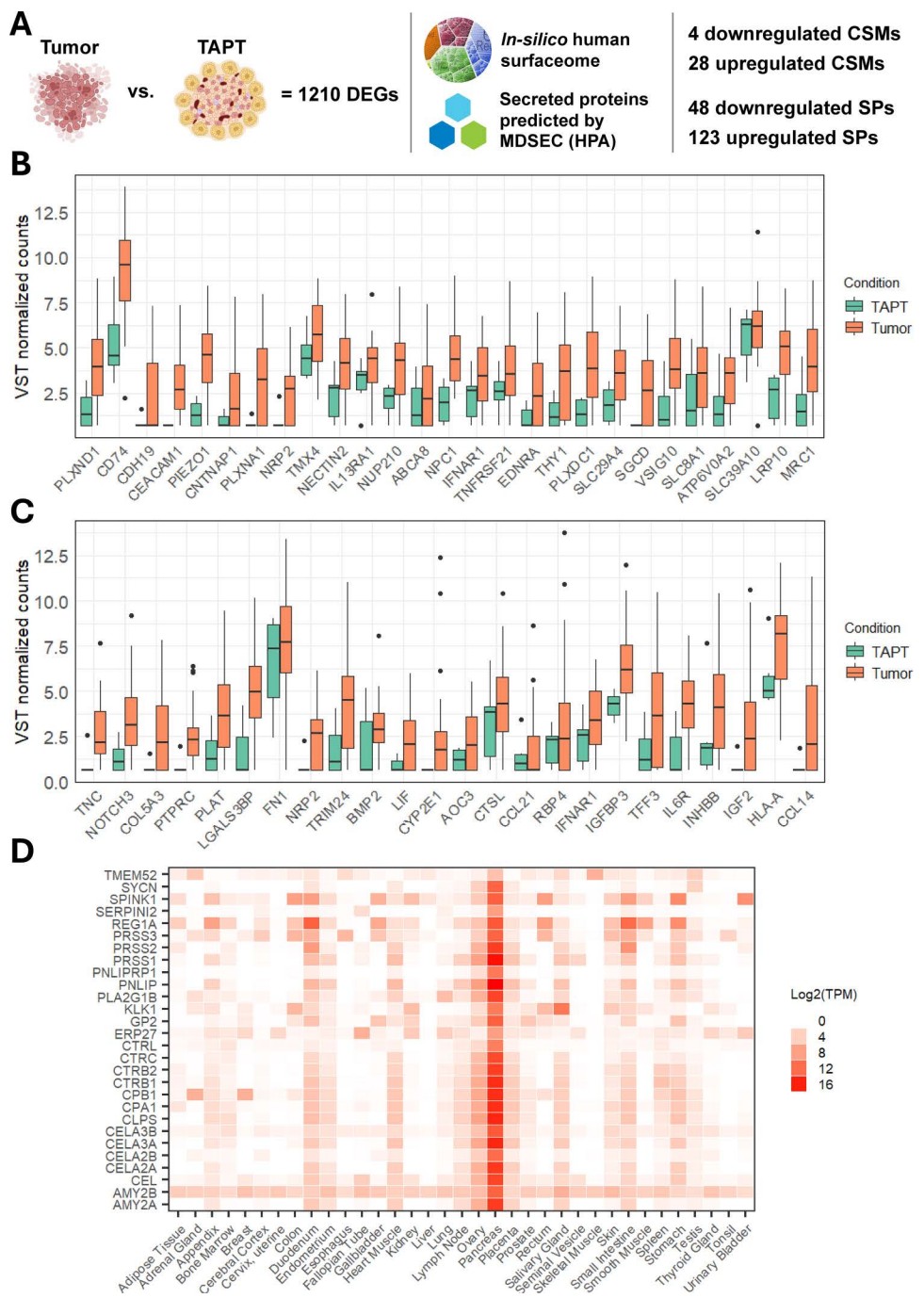

**Fig 5. Cell surface markers (CSMs) and secretome-related proteins (SPs) discovered in the differential gene expression analysis results data-set comparing tumors and tumor-adjacent pancreatic tissues (TAPT).** A – The search of differentially expressed genes in silico human surfaceome and Human Protein Atlas (HPA) databases. B – Boxplots of 28 upregulated CSMs in tumor samples. The Y-axis represents variance-stabilizing transformed normalized counts. X-axis represents gene symbols according to HUGO gene nomenclature. C – Boxplots of 24 upregulated cancer secretome associated SPs in tumor samples. The Y-axis represents variance-stabilizing transformed normalized counts. D – Heatmap representing calculated transcripts per million (TPM) expression values for the identified SPs that were enriched for pancreatic tissue.

Regarding the secretome proteins (SP), we discovered that 171 DEGs (S6 Table) had an entry in the Human Protein Atlas (HPA) list of secreted proteins predicted by MDSEC (a majority decision-based method for secreted proteins). Of these, 48 were upregulated and 123 were downregulated (Fig 5A). Among these, there were 24 upregulated genes related to the cancer secretome according to the HPA database (Fig 5C). To determine whether the 171 genes reflect a specific tissue type, we carried out tissue type enrichment analysis, which revealed 28 genes specific to pancreatic tissue (Fig 5D), and all 28 had downregulated expressions. Interestingly, all 28 genes were downregulated within PanNETs compared to TAPTs. This is not entirely unexpected, as according to the STRING database, the majority of these genes are related to protein/fat digestion and absorption, as well as pancreatic secretion.

The expression of the 27 upregulated CSMs and 24 upregulated SPs was further evaluated in two publicly available PanNET datasets from Gene Expression Omnibus (Accession no. GSE98894 and GSE116356). Here it can be observed (S5 Fig) that the expression of these markers is indeed present within PanNET tissues. Unsurprisingly, the expression of *RBP4* and *IGF2* is the highest in liver metastasis as these proteins are primarily produced and secreted by liver.

## Discussion

In this study, we aimed to expand on the knowledge in the field of PanNET transcriptomics by analyzing tissue samples from a cohort of 30 PanNET patients. The cohort represented four populations (Latvia, Spain, Greece, and Slovakia). Our goal was to identify alterations in gene expression profiles, overrepresented molecular pathways and potential markers for developing non-invasive testing assays. As indicated by our results, we identified intriguing observations that align with the current understanding of tumor biology, including PanNETs (PI3K, Wnt, and MAPK signaling) [39] as well as novel findings in the context of PanNETs—specifically, the dysregulation of the neddylation pathway.

Overall, the DGE analysis comparing 30 PanNETs to six TAPT samples yielded a significant number of dysregulated genes (1210 total; 1041 upregulated and 169 downregulated). Here a multitude of previously well-established oncogenic pathways [41–44] were overrepresented among the list of 1210 DEGs. Notably, Wnt and MAPK signaling processes were among the most overrepresented Reactome terms (according to FDR value) (Fig 3B). Additionally, we observed terms related to PI3K signaling (S3 Table), which is one of the most studied pathways in the context of PanNETs [45].

Alterations in Wnt signaling pathways have been noted previously in PanNETs [46] as the Wnt signaling is involved in the development of pancreatic islets [47]. The impact of alterations in Wnt signaling is for the most part is unclear in PanNETs [46]. Nevertheless there are studies suggesting that the Wnt signaling may play a role in PanNET development as there is evidence pointing at alterations in β-catenin signaling [37], β-catenin independent, and TCF dependent signaling [36].

Disruption of MAPK signaling has also been previously observed in NF-PanNETs specifically regarding *RASSF1A* gene, a proposed tumor suppressor that encodes protein similar to classical RAS effectors. In these studies a silencing of this gene was detected due to promoter hypermethylation [48,49]. Another, recently published *in vitro* study, found that a targeted inhibition of the *FASN* gene (which was found to be overexpressed in the study) via Orlistat treatment can impact MAPK signaling, leading to ferroptosis [50]. In our data, we do not observe dysregulation of the *FASN* or *RASSF1* genes, nor interactions with these genes in the list of MAPK signaling-related DEGs from STRING analysis (S3 Fig). Such discrepancy between results could be attributed to the overall intra-patient heterogeneity of PanNETs [51].

Everolimus has been FDA-approved treatment for advanced PanNETs for well over 10 years [52]. The use of Everolimus is somewhat limited by the amount of unsatisfactory response rates [53–55]. One key inducer of PI3K signaling is insulin-like growth factor signaling (IGF). It has been shown that upregulation of IGF1 can induce treatment resistance in everolimus-responsive PanNETs [54]. Our data indicate that all tumors express *IGF1* and IGF1 receptor (*IGF1R*); however, we do not observe any upregulation of these genes compared to TAPTs. Nevertheless, we do see upregulation in *IGF2* and *IGFBP3* (S2 Table). Similarly to IGF1, these factors can activate PI3K signaling through IGF1R. It has been proposed that cancer-associated fibroblasts are one of the sources of IGF2 and IGFBPs in the tumor microenvironment

[56–58]. Our previous findings also showed a substantial upregulation of *IGFBP3* in both α-SMA-expressing stromal cells and tumor cells; unfortunately, expression levels of *IGF2* could not be evaluated in that study as it was not included within the analysis panel [36].

Looking at the overall results from a broader perspective, we can observe that the altered genes related to Wnt, MAPK, and PI3K signaling physically interact with commonly known tumor suppressor genes (TSGs) such as *TP53* and *PTEN*, as well as oncogenes like *KRAS*, *HRAS*, *NRAS*, *BRAF*, and *EGFR* (Figs S2, S3, and S4). Mutations and loss of *PTEN* expression have been observed in PanNETs [13,59]. On the other hand, alterations in *KRAS*, *BRAF*, *EGFR*, and *TP53* have rarely, if ever, been observed in primary tumors of PanNETs [14,60]. These genes were also not differentially regulated in PanNETs according to our results. Exceptions are the metastases of PanNETs, where alterations in these genes have been observed [61]. Additionally, these genes are among the most frequently altered genes in pancreatic cancer [62–65]. This indicates that direct loss or gain-of-function alterations in these genes are more common in aggressive tumors, such as pancreatic adenocarcinomas, rather than in neuroendocrine tumors. This raises a question. Perhaps in PanNETs, the major components of Wnt, MAPK, and PI3K signaling, along with commonly known oncogenes and tumor suppressor genes (TSGs), could be activated or suppressed by the interacting DEGs, as shown in our results (Figs S2, S3, and S4). Particularly interesting observations in our results are the overexpressed proteasomal complex genes: *PSMA4*, *PSMB9*, *PSMC1*, *PSMC2*, *PSMD9*, and *PSME2*, which are associated with all three signaling pathways (Wnt, MAPK, PI3K). Dysregulated proteasomes have been proposed as one of the drivers of tumorigenesis, as they can disrupt the normal degradation of proteins involved in cell cycle regulation, proliferation, and survival [66,67]. Moreover, one of the recently discovered regulatory mechanisms of proteasome activity is neddylation—a process that involves the ligation of a protein similar to ubiquitin, NEDD8, to mitigate the effects of proteotoxic stress [68]. The two main substrates of neddylation are cullin and non-cullin proteins. Neddylation of cullin family proteins results in the activation of cullin RING ligases – the largest known class of ubiquitin ligases; accordingly, cullin family protein neddylation acts as one of the control mechanisms for protein ubiquitination itself [69]. In our list of DEGs, neddylation was the most overrepresented pathway. Although, based on these results, it is not possible to precisely define whether the neddylation is disrupted within PanNETs, it is evident here that the processes related proteasomal activity regulated by neddylation are altered (S1 Fig) in PanNETs warranting further functional investigation.

Regarding the results of CSMs and SPs. Here, among the 27 upregulated CSMs, we discovered several genes related to malignancies, including Plexins A-1/D-1 and Plexin domain-containing protein 1 (*PLXNA1*, *PLXND1*, and *PLXDC1*) and Cadherin-19 (*CDH19*). We also observed an overexpression of Neuropilin-2 (*NRP2*) in tumor tissue. *NRP2* is known to be an an integral part of the VEGF signaling system [70], one of the key signaling pathways in PanNETs [71]. This is interesting as in a different PanNET study regarding *NRP2,* it was found that NRP2 protein is expressed in circulating tumor cells sampled from NET patients. However, its capacity to discriminate between NETs and healthy controls was somewhat dubious [72].

Other notable NET-related CSM and SP genes include previously discussed *IGF-2* and *IGFBP3* genes and genes related to the stromal component of the tumor (*COL5A3*, *FN1*, and *NOTCH3*). The upregulation of some of these genes was already reported in our previous study on PanNETs. Here, the expression of *COL5A3*, *FN1*, and *NOTCH3* was more specific to α-SMA-positive stromal cells rather than tumor cells, while *IGFBP3* was overexpressed in both tumor and stroma in comparison to adjacent non-tumor tissues [36]. There are also reports that IGFBPs, including *IGFBP3*, are overexpressed in gliomas compared to non-tumor brain tissue and with higher levels of expression observed in high-grade gliomas [73]. An interesting aspect is the overexpression of *IGF-2* in PanNETs, as this gene not only plays a role in the activation of the PI3K axis [57], but has also been proposed as one of the key drivers of tumorigenesis in ileal NETs [74]. Exuberant levels of IGF-2 secreted by the tumor tend to cause hypoglycemia that mimics hypoglycemia caused by insulin-secreting PanNETs [75]. Therefore, in clinical settings, circulating IGF-2 is only measured when there is a suspicion of non-islet cell tumor-induced hypoglycemia. However, it would be interesting to see a larger patient cohort study that assesses the levels of IGF-1/2 and IGFBPs in the plasma of PanNET patients aswell. Perhaps the levels of IGF

and their binding proteins could be elevated in patients harboring NF-PanNETs to an extent that would not be enough to induce hypoglycemia but sufficient enough to indicate the presence of a NET.

Lastly, we also investigated the transcriptomic changes across different grades of PanNETs. Similar to previous studies [36,76] the highest number of DEGs can be observed in the G3 vs. G1 comparison, highlighting the differences between G3 tumors and G2/G1 tumors. In G3 vs. G2/G1 comparisons, we observed that G3 tumors have a significantly lower expression of *NCAM1*, a marker of neuroendocrine differentiation [77]. This gene was also downregulated in G3 vs. G1 comparison indicating that there is some degree of dedifferentiation in G3 tumors. Furthermore, compared to G1 tumors the most upregulated gene in G3 tumors was *LCN2*. It encodes a known oncogene also referred to as 24p3 [78]. The upregulation of *LCN2* has been observed in pancreatic cancer [79]. Upon further investigation, it was revealed that *LCN2* could serve as a potential therapeutic target, as depletion of *LCN2* in a pancreatic cancer mouse model resulted in increased survival [79]. Interestingly, in previous RNA-seq based PanNET study this gene was also overexpressed in G3 PanNETs compared to G1 PanNETs [76]. Amongst the list of upregulated genes in G3 vs. G1 tumors (S4 Table) there were three other genes with oncogenic properties – *PLEKHG2*, *MYDGF*, and *FOXQ1*. Upregulation of *PLEKHG2* has been observed in lung and pancreatic cancer where it correlated with poor survival [80,81]. *MYDGF* encodes a growth factor with anti-apoptotic and angiogenesis-stimulating properties. It has recently gained significant attention in cancer research, as it has been observed to be upregulated in multiple malignancies [82]. Lastly, *FOXQ1* is a well-known regulator of Wnt signaling. In cancer, the upregulation of *FOXQ1* is associated with significantly worse survival due to its ability to promote epithelial-to-mesenchymal transition, thereby allowing tumor cells to metastasize [83].

There were several limitations in this study. The first limitation – low number of G3 tumor cases. This can be attributed to the general rarity of G3 PanNETs [84]. The low number of G3 cases has consistently been a limiting factor in similar omics-based studies [36,76,84]. The second limitation can be attributed to the low availability of TAPT samples as in this study only six patients had available TAPT tissues. In a clinical setting, the tumor resection margins can significantly vary from patient to patient. As such the non-tumor tissue may be enough for robust histopathological evaluation but not in sufficient quantities for RNA-seq analysis. The third limitation can be attributed to the histology of used adjacent non-tumor tissues that included both exocrine and endocrine parts of the pancreas. It is known that the largest component (around 95%) of the normal pancreas is the exocrine component [85]. However, PanNETs originate from the endocrine component, specifically from α and β islet cells, as indicated by their epigenetic signatures [86]. Accordingly, the due to cell type mixing [87] of exocrine and endocrine pancreas cells a percentage of identified DEGs between non-tumor tissue and PanNETs are related to exocrine pancreas functions.

## Conclusions

In conclusion, we performed a comprehensive profiling of the PanNET tissue transcriptome and demonstrated that the dysregulated genes correspond to known oncogenic signaling pathways, including Wnt, MAPK, and PI3K. These pathways are likely further affected by the observed dysregulation of genes related to proteasomal activity, which is regulated by neddylation-related processes. Furthermore, we report that PanNETs also overexpress *NRP2*, *IGF2*, and *IGFBP3*-genes encoding proteins that can be assayed in blood plasma. These findings suggest that, in the future, these proteins could potentially be used for the non-invasive detection of PanNETs via liquid biopsy.

## Supporting information

**S1 Fig. Protein-protein interaction network (PPI) displaying both functional and physical protein-protein interactions of differentially expressed genes related to Neddylation Reactiome pathway.** To further understand the cellular components and processes represented by these genes a k-means clustering analysis was performed obtaining two clusters. Gene Ontology (GO) cellular components and processes represented by each of the clusters are detailed on the right side of the image. The network was generated using STRING database (v12.0) and on STRING website. Line

thickness between the nodes represents confidence scores of interaction which were calculated using following channels: textmining, experiments, databases, co-expression, neighborhood, gene fusion, co-occurrence.
(TIF)

**S2 Fig. Protein-protein interaction network (PPI) displaying the physical protein-protein interactions of differentially expressed (tumor vs. non-tumor) genes related to β-catenin independent WNT signaling and TCF-dependent signaling in response to WNT Reactome pathways.** The network also includes 20 closest interacting neighbors (grey nodes) that were not differentially expressed. The graph was made in Cytoscape (v3.10.2) using data from STRING database (v12.0); channels used to calculate interaction confidence scores: textmining, experiments, databases, co-expression, neighborhood, gene fusion, co-occurrence.
(TIF)

**S3 Fig. Protein-protein interaction network (PPI) displaying the physical protein-protein interactions of differentially expressed (tumor vs. non-tumor) genes related to signaling by RAF1 Mutants, MAPK1/MAPK3 signaling, and RAF/MAP Kinase Cascade Reactome pathways.** The network also includes 20 closest interacting neighbors (grey nodes) that were not differentially expressed. The graph was made in Cytoscape (v3.10.2) using data from STRING database (v12.0); channels used to calculate interaction confidence scores: textmining, experiments, databases, co-expression, neighborhood, gene fusion, co-occurrence.
(TIF)

**S4 Fig. Protein-protein interaction network (PPI) displaying the physical protein-protein interactions of differentially expressed (tumor vs. non-tumor) genes related to PI3K/AKT Signaling in Cancer, PIP3 activates AKT signaling, and PTEN regulation Reactome pathways.** The network also includes 20 closest interacting neighbors (grey nodes) that were not differentially expressed. The graph was made in Cytoscape (v3.10.2) using data from STRING database (v12.0); channels used to calculate interaction confidence scores: textmining, experiments, databases, co-expression, neighborhood, gene fusion, co-occurrence.
(TIF)

**S5 Fig. Expression of upregulated cell surface markers (CSMs) (panel A) and secretome proteins (SPs) (Panel B) in our data and publicly available datasets (GSE98894 and GSE116356) from Gene Expression Omnibus.** Y-axis – variance stabilizing transformed normalized counts, X-axis – gene symbols. Of the total 27 CSMs *NECTIN2* is missing as it was not found in count data of the publicly available datasets. Of the total 24 SPs genes *TNC* and *HLA-A* are missing as they were not found in count data of the publicly available datasets.
(TIF)

**S1 Table. Patient clinical data.**
(XLSX)

**S2 Table. Results of differential expression analysis between pancreatic neuroendocrine tumor tissues vs tumor adjacent pancreatic tissues.**
(XLSX)

**S3 Table. Results of active subnetwork-oriented pathway enrichment analysis.**
(XLSX)

**S4 Table. Results of differential expression analysis between different grades of pancreatic neuroendocrine tumors.**
(XLSX)

**S5 Table. A list of DEGs from tumor vs. tumor-adjacent tissue comparisons that are labeled as cell surface markers according to the in-silico human surfaceome database.**
(XLSX)

**S6 Table. A list of DEGs from tumor vs. tumor-adjacent tissue comparisons that are labeled as secretome proteins according to the human protein atlas (HPA).** List sourced from – secretome proteins predicted by MDSEC (majority decision-based method for secreted proteins) section.
(XLSX)

**S1 Text. R code that was used to perform the included statistical analyses.** The ffile includes code and comments regarding differential expression analysis, pathway enrichment analysis, and tissue enrichment analysis.
(DOCX)

## Acknowledgments

The authors acknowledge the Latvian Biomedical Research and Study Centre as well as the National Biobank – Genome Database of the Latvian Population for providing samples, data, infrastructure, and support. We also want to acknowledge the patients and the BioBank Hospital Ramón y Cajal-IRYCIS (PT17/0015/0010) integrated within the Spanish National Biobanks Network for its collaboration.

## Author contributions

**Conceptualization:** Helvijs Niedra, Olesja Rogoza, Raitis Peculis, Ignacio Ruz-Caracuel, Julie Earl, Bozena Smolkova, Agapi Kataki, Vita Rovite.

**Data curation:** Rihards Saksis, Aija Gerina, Sofija Vilisova, Natalja Senterjakova, Aldis Pukitis, Ignacio Ruz-Caracuel, Georgina Kolnikova, Peter Dubovan, Miroslav Tomas, Peter Makovicky, Maria Urbanova, Eythimios Koniaris, Ioanna Aggelioudaki.

**Formal analysis:** Helvijs Niedra.

**Funding acquisition:** Julie Earl, Bozena Smolkova, Agapi Kataki, Vita Rovite.

**Investigation:** Helvijs Niedra, Olesja Rogoza, Raitis Peculis.

**Methodology:** Helvijs Niedra, Olesja Rogoza, Rihards Saksis, Raitis Peculis, Anzela Halilova, Ignacio Ruz-Caracuel.

**Project administration:** Bozena Smolkova, Agapi Kataki.

**Supervision:** Julie Earl, Bozena Smolkova, Agapi Kataki, Vita Rovite.

**Validation:** Rihards Saksis, Raitis Peculis.

**Writing – original draft:** Helvijs Niedra, Olesja Rogoza.

**Writing – review & editing:** Helvijs Niedra, Olesja Rogoza, Raitis Peculis, Vita Rovite.

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
