## [Decision Letter · Decision Letter 0]

Dear Dr. Niedra,

Thank you for submitting your manuscript to PLOS ONE. After careful consideration, we feel that it has merit but does not fully meet PLOS ONE’s publication criteria as it currently stands. Therefore, we invite you to submit a revised version of the manuscript that addresses the points raised during the review process.

We look forward to receiving your revised manuscript.

Kind regards,

Li Shen

Academic Editor

PLOS ONE

Journal Requirements:

This research was supported by the European Regional Development Fund project “Establishing an algorithm for the early diagnosis and follow-up of patients with pancreatic neuroendocrine tumors – NExT” project (1.1.1.5/ERANET/20/03) and the EUs Horizon 2020 research and innovation program (grant No. 857381/VISION). Partners from Slovakia were supported by the Slovak Academy of Sciences, Slovakia (NExT-0711) and the Slovak Research and Development Agency (APVV-21-0197, APVV-20-0143). Partners from Greece received funding from the TRANSCAN-2 program ERA-NET JTC 2017 “Translational research on rare cancers” within the project NExT (NKUA): Ministry of Education, Research and Religious Affairs - General Secretariat for Research and Technology, Greece (Τ9ЕРА3-00012).

Lastly, during the manuscript preparation and review process, Rihards Saksis was supported by the project "Strengthening of the capacity of doctoral studies at the University of Latvia within the framework of the new doctoral model”, identification No.8.2.2.0/20/I/006. During the study Helvijs Niedra was supported within the framework of the European Union’s Recovery and Resilience Mechanism project No.5.2.1.1.i.0/2/24/I/CFLA/001 "Consolidation of the Latvian Institute of Organic Synthesis and the Latvian Biomedical Research and Study Centre" (Grant subsection identifier: ANM_K_DG_07).  

Reviewers' comments:

Reviewer's Responses to Questions

**Comments to the Author**

1. Is the manuscript technically sound, and do the data support the conclusions?

Reviewer #1: Yes

Reviewer #2: Yes

Reviewer #3: Partly

2. Has the statistical analysis been performed appropriately and rigorously?

Reviewer #1: Yes

Reviewer #2: No

Reviewer #3: Yes

3. Have the authors made all data underlying the findings in their manuscript fully available?

Reviewer #1: Yes

Reviewer #2: Yes

Reviewer #3: Yes

4. Is the manuscript presented in an intelligible fashion and written in standard English?

Reviewer #1: Yes

Reviewer #2: Yes

Reviewer #3: Yes

Reviewer #1: The research is valuable and well-organized, though certain aspects need refinement for clarity and completeness.

Detailed Comments

Abstract

Line 5:

Line 12: Include the total number of differentially expressed genes (DEGs) and specify the significance thresholds used for their identification to strengthen the abstract.

Introduction

3. Line 20: Provide updated statistics on cancer in general as well as this cancer type prevalence, including survival rates, to highlight the critical need for prognostic biomarkers. Cite “Cancer statistics, 2024, 2024”. Then give intro in cancer therapy in general, cite NIH paper “Cancer treatments: Past, present, and future, 2024” (PMID: 38909530)for more information.

Line 40: Expand on the clinical and biological challenges posed by PanNETs, such as their asymptomatic nature and high metastatic potential.

4. Line 65: Include a discussion on the potential implications of transcriptomic profiling for personalized medicine in PanNETs.

5. Line 75: Cite recent studies addressing molecular markers and pathways in PanNETs, emphasizing areas of agreement or contradiction with the current study.

Materials and Methods

7. Line 150: Provide additional details on the preprocessing of RNA-seq data, including quality control metrics and read trimming thresholds.

8. Line 170: Discuss the justification for using the STRING and Reactome databases in the pathway enrichment analyses.

Results

9. Line 200: Highlight the biological relevance of the identified overrepresented pathways, such as their roles in PanNET tumorigenesis and progression.

10. Line 240: Include more granular details on the differentially expressed genes (DEGs) across tumor grades, particularly focusing on unique markers for G3 tumors.

11. Line 280: Discuss why certain genes (e.g., IGF2, IGFBP3) were chosen as potential biomarkers and their implications for liquid biopsy development. Mention previous study reporting IGFBP as glioma biomarkers: “A bioinformatic study of IGFBPs in glioma regarding their diagnostic, prognostic, and therapeutic prediction value, 2023”

Discussion

12. Line 350: Provide a critical comparison of the findings with prior studies on PanNET-related pathways, particularly regarding WNT and MAPK signaling. Provide a more detailed context for the relevance of the identified pathways (e.g., WNT, MAPK, PI3K) in tumorigenesis. Mention recent studies reporting association of these pathway in cancer. Such as“Isocucurbitacin B inhibits glioma growth through PI3K/AKT pathways and increases glioma sensitivity to TMZ by inhibiting hsa-mir-1286a, 2024” “MAPK signaling pathway-based glioma subtypes, machine-learning risk model, and key hub proteins identification, 2023,PI3K/AKT/mTOR signaling in gastric cancer: Epigenetics and beyond, 2020”

13. Line 400: Address potential limitations of the study, such as the low sample size for G3 tumors, and propose strategies for addressing these in future research. Discuss the bias from transcriptional data, refer to “Genetic expression in cancer research: Challenges and complexity, 2024” and “Technical and Biological Biases in Bulk Transcriptomic Data Mining for Cancer Research, 2025”

14. Line 450: Suggest how the identified biomarkers could be integrated into clinical workflows for PanNET diagnosis and management.

Suggest future studies that could validate these findings in patient-derived xenograft models or larger cohorts. Previous studies using xenograft models of cancer should be mentioned, such as “Comparing volatile and intravenous anesthetics in a mouse model of breast cancer metastasis, 2018”

Figures and Tables

15. Figure 2: Annotate the heatmaps with additional labels or legends to clarify the significance of the identified DEGs.

16. Figure 5: Include additional context for the significance of cell surface markers (CSMs) and secretome proteins (SPs) in the clinical management of PanNETs.

Reviewer #2: Dear Authors,

This article explores a clinically significant topic by utilizing transcriptomic profiling to identify potential biomarkers for PanNETs. The study is enriched by the inclusion of pathway analysis and the identification of non-invasive biomarkers, which add substantial depth to the research. The emphasis on secretome proteins and liquid biopsy markers is particularly noteworthy, as it aligns with current advancements in non-invasive cancer diagnostics and enhances the translational relevance of the findings. However, I have a few questions that need to be addressed.

1. The title appears somewhat complex; please rephrase the title while retaining the focus on pathways and biomarkers.

2. There seems to be an imbalance in the number of samples between PanNETs and TAPTs. Do you think it may affect the statistical tests? Please justify it.

3. Could you elaborate on the statistical methodologies used for clustering analysis? Specifically, how were significant clusters and pathways determined?

Thanks

Reviewer #3: In this manuscript, authors have performed a transcriptomic analysis of pancreatic neuroendocrine tumor tissues. Here are a few concerns regarding the manuscript-

1. The first major issue with the study is novelty. Similar studies have been performed earlier. Authors should describe a strong rationale for this study.

2. Another issue with the manuscript is lack of details for each figure panel. For example, in Fig. 2, authors write 1 line for all the panels in this figure and mention that a number of genes were differently expressed. Without calling out each panel in the text, it will be difficult to follow the manuscript.

3. The authors have discussed the results in detail and tried to make sense of existing data. Some of this information should be incorporated into results section as it will make better sense there.

4. Authors mention that grade 3 (G3) tumor is more proliferative in comparison to other lower grades, however, they don’t find genes linked proliferation upregulated in G3 tumor samples. Could this be due to miss-classification. G3 and G2 look transcriptionally very similar.

5. Authors have not performed any experiment to validate their findings, this makes it underwhelming.

**Do you want your identity to be public for this peer review?** For information about this choice, including consent withdrawal, please see our Privacy Policy

Reviewer #1: No

Reviewer #2: No

Reviewer #3: No

---

## [Author Response · Author response to Decision Letter 1]

20 Mar 2025

Reviewer 1

Comment 1

Include the total number of differentially expressed genes (DEGs) and specify the significance thresholds used for their identification to strengthen the abstract. Thank you for suggesting to add statistical thresholds. We updated the abstract. Please see added lines (27 - 29) in tracked changes.

Response 1

Thank you for suggesting to add statistical thresholds. We updated the abstract. Please see added lines (27 - 29) in tracked changes.

Comment 2

Line 20: Provide updated statistics on cancer in general as well as this cancer type prevalence, including survival rates, to highlight the critical need for prognostic biomarkers. Cite “Cancer statistics, 2024, 2024”. Then give intro in cancer therapy in general, cite NIH paper “Cancer treatments: Past, present, and future, 2024” (PMID: 38909530) for more information.

Response 2

Thank you for the comment within "Introduction" section line 51, we have briefly stated the incidence of PaNETs and treatment information. According to your suggestion we have expanded this in line (lines 53-54 in tracked changes). We added more details regarding overall survival statistics of patients with PanNETs. Regarding article PMID:38909530. It is a valuable review, however, it includes no information regarding neuroendocrine tumors which are rather different from "classical" cancers in terms of differentiation and clinical aggressiveness. Regardless, we still included the article within the literature review as it discusses that surgery remains as the primary treatment option for most forms of cancer. This is also the same for neuroendocrine tumors. Regarding the treatment of PanNETs, we have already briefly discussed this within lines 51 - 61.

Comment 3

Line 40: Expand on the clinical and biological challenges posed by PanNETs, such as their asymptomatic nature and high metastatic potential.

Response 3

Thank you for the comment. The metastatic risk is discussed in lines 52 - 53 where we found in a review by Das S and Dasari A that metastatic involvement is present in between of 21 and 65% of diagnosed cases. We've briefly discussed challenges regarding current serum biomarkers for diagnostics (chromogranin A, neuron-specific enolase, and pancreatic peptide) in lines 62-63. Lastly, we've also highlighted their asymptomatic nature which mainly concerns non-functioning tumors, please see lines 65 - 67.

Comment 4

Line 65: Include a discussion on the potential implications of transcriptomic profiling for personalized medicine in PanNETs.

Response 4

Line 65: Include a discussion on the potential implications of transcriptomic profiling for personalized medicine in PanNETs.

Comment 5

Line 75: Cite recent studies addressing molecular markers and pathways in PanNETs, emphasizing areas of agreement or contradiction with the current study.

Response 5

Thank you for the comment. We have discussed this in lines 68 - 79, highlighting, the currently most significant molecular markers ATRX/DAXX and PTEN, TSC2, PIK3CA, and MEN1.

Comment 6

Line 150: Provide additional details on the preprocessing of RNA-seq data, including quality control metrics and read trimming thresholds.

Response 6

We have provided the details regarding quality metrics in Materials and methods, "data analysis" subsection (lines 130 - 133).

Comment 7

Line 170: Discuss the justification for using the STRING and Reactome databases in the pathway enrichment analyses.

Response 7

Thank you for the suggestion. We chose the STRING database as it is compatible with PathfindR active subnetwork-based enrichment analysis. Accordingly, STRING database was used to generate the PPI interaction network which was segregated into active subnetworks by PathfindR tool. After this the genes within identified subnetworks were used as input to perform pathway enrichment analysis using Reactome database. We chose specifically this database as according to methodological study by Gable et al. (PMID: 36088548) it provided the highest overall performance of manually curated annotation systems for human genome related studies. Accordingly, we included this information manuscript. Please see tracked changes in lines 148 - 152.

Comment 8

Line 200: Highlight the biological relevance of the identified overrepresented pathways, such as their roles in PanNET tumorigenesis and progression.

Response 8

Thank you for the comment. We have updated results section highlighting the biological relevance of identified overrepresented pathways. This was done by moving some parts of the discussion to results section as such change was also requested by another reviewer. Please see tracked changes in lines 234 - 237, 240 - 252, and 266 - 275.

Comment 9

Line 240: Include more granular details on the differentially expressed genes (DEGs) across tumor grades, particularly focusing on unique markers for G3 tumors.

Reponse 9

Thank you for the comment In section "Transcriptomic changes across different grades of PanNETs" we have added some details regarding NCAM1 gene expression in G3 tumors compared to G2 and G1 tumors. Please see tracked changes in lines 289 - 293. NCAM1 is known to be another marker of Neuroendocrine differentiation in NETs (particularly lung NETs) and we can observe that the expression of NCAM1 is significantly downregulated in G3 tumors compared to G2 and G1. This is indicates that G3 tumors, while still histologically showing features of well-differentiated tumors, are slightly less differentiated than G2 and G1 NETs in terms of NCAM1 expression.

Comment 10

Line 280:Discuss why certain genes (e.g., IGF2, IGFBP3) were chosen as potential biomarkers and their implications for liquid biopsy development. Mention previous study reporting IGFBP as glioma biomarkers: “A bioinformatic study of IGFBPs in glioma regarding their diagnostic, prognostic, and therapeutic prediction value, 2023”

Response 10

Thank you for the comment. We've discussed to some extent why we focused on specifically IGF2 and IGFBPs in lines 388 - 399. However, it is interesting to see that the involvement of IGFBPs have been also assessed in gliomas with interesting results. This further provides another layer of evidence that IGF axis does play a significant role in carcinogenesis. Accordingly, we've included article “A bioinformatic study of IGFBPs in glioma regarding their diagnostic, prognostic, and therapeutic prediction value, 2023” within our manuscript. Please see tracked changes in lines 438 - 440.

Comment 11

Line 350: Provide a critical comparison of the findings with prior studies on PanNET-related pathways, particularly regarding WNT and MAPK signaling. Provide a more detailed context for the relevance of the identified pathways (e.g., WNT, MAPK, PI3K) in tumorigenesis. Mention recent studies reporting association of these pathway in cancer. Such as“Isocucurbitacin B inhibits glioma growth through PI3K/AKT pathways and increases glioma sensitivity to TMZ by inhibiting hsa-mir-1286a, 2024” “MAPK signaling pathway-based glioma subtypes, machine-learning risk model, and key hub proteins identification, 2023,PI3K/AKT/mTOR signaling in gastric cancer: Epigenetics and beyond, 2020”

Response 11

Thank you for the comment. We have provided some comparison of our results regarding WNT and MAPK signaling in PanNETs compared to other studies. Please see lines 241-252 and 366 - 370 (wnt); lines 248 - 252 and 379- 387 (MAPK signaling); lines 266 - 275 and 388 - 399 (PI3K and mTOR signaling). The studies that are provided within the comment are related to gliomas and gastric cancer which are entire different tumors compared to pancreatic neuroendocrine tumors. While we would like to primarily focus on PanNET studies for comparing our results to literature. But we chose to include the studies: “MAPK signaling pathway-based glioma subtypes, machine-learning risk model, and key hub proteins identification, 2023" and "PI3K/AKT/mTOR signaling in gastric cancer: Epigenetics and beyond, 2020” within introductory section of the discussion to generally excel the fact that these are oncogenic pathways that are dysregulated in many types of cancers. Changes were made in line 361.

Comment 12

Line 400: Address potential limitations of the study, such as the low sample size for G3 tumors, and propose strategies for addressing these in future research. Discuss the bias from transcriptional data, refer to “Genetic expression in cancer research: Challenges and complexity, 2024” and “Technical and Biological Biases in Bulk Transcriptomic Data Mining for Cancer Research, 2025”

Response 12

Thank you for the comment. We have already discussed the low sample size of G3 tumors as one of the primary limitations (Please see lines 450-462). The article "Genetic expression in cancer research: Challenges and complexity, 2024" is indeed interesting as it discusses common pitfalls of bulk RNA-seq studies. We cited this study when discussing the limitations of using non-dissected adjacent pancreatic tissue. This type of tissue contains a mix of both exocrine and endocrine pancreas cells and due to cell type mixing not all DEGs represent changes in tumor compared to cells of origin (endocrine pancreas). Some of the DEGs are related to digestive functions of exocrine pancreas, such as CPA genes which are downregulated in our tumor samples.

Comment 13

Line 450: Suggest how the identified biomarkers could be integrated into clinical workflows for PanNET diagnosis and management.

Suggest future studies that could validate these findings in patient-derived xenograft models or larger cohorts. Previous studies using xenograft models of cancer should be mentioned, such as “Comparing volatile and intravenous anesthetics in a mouse model of breast cancer metastasis, 2018”

Response 13

Thank you for the comment. In context of PanNETs we've raised these suggestions in discussion section: lines 388 - 399 and 433 -449. However, we do not think it would be appropriate to discuss the article "Comparing volatile and intravenous anesthetics in a mouse model of breast cancer metastasis, 2018" as it is a conference abstract and it is out context for this study.

Comment 14

Annotate the heatmaps with additional labels or legends to clarify the significance of the identified DEGs.

Response 14

Thank you for the comment. The heatmaps in Figures 2 and 4 represent top 25 upregulated and downregulated genes with FDR value < 0.05. In Figure 2 D heatmap we've added 2 annotations: 1) sample type which describes whether the sample is tumor tissue or tumor adjacent pancreatic tissue (TAPT); 2) tumor grade which describes what is the histological grade off the tumor sample (G1 - grade 1, G2 - grade 3, G3 - grade 3).

Comment 15

Include additional context for the significance of cell surface markers (CSMs) and secretome proteins (SPs) in the clinical management of PanNETs.

Response 15

We have provided these details in discussion section. Please see lines 425 - 432 and 433 - 449

Reviewer 2

Comment 1

1. The title appears somewhat complex; please rephrase the title while retaining the focus on pathways and biomarkers.

Response 1

Thank you for the suggestion. We made some minor changes to our title to make it less complex while retaining the focus on pathways and biomarkers. This would be our new title "Transcriptomic profiling of pancreatic neuroendocrine tumors: dysregulation of WNT, MAPK, PI3K, neddylation pathways and potential non-invasive biomarkers"

Comment 2

2. There seems to be an imbalance in the number of samples between PanNETs and TAPTs. Do you think it may affect the statistical tests? Please justify it.

Response 2

The difference in number of tumor samples and tumor adjacent pancreatic tissues (TAPTs) was primarily due to tissue availability as most of the resected tumor specimens did not contain enough non-tumor tissue. For future studies this could be alleviated by using microdissection techniques to acquire enough tumor free material. Unfortunately, six TAPT samples was all we could collect for this study. This, of course, may impact the overall statistical power of this study. To somewhat alleviate this issue we employed stringent filtering of differential expression analysis results performing Log2 Fold Change value shrinkage using "apeglm" method developed by Anqi Zhu et al. (PMID: 30395178). By applying this method, we filtered out genes that came up as differentially expressed because one of the control samples had abnormally high or low read counts of that specific gene resulting in overestimated fold changes.

Comment 3

3. Could you elaborate on the statistical methodologies used for clustering analysis? Specifically, how were significant clusters and pathways determined?

Response 3

Thank you for the comment. To carry out pathway enrichment analysis we used PathfindR package (v2.4.1) developed by Uglen et. al (Published paper PMID: 31608109). Here the enrichment analysis is based on active protein-protein interaction subnetworks. Accordingly, this method first generates a protein-protein interaction network using various databases, in our case we resorted to STRING database. Following this, the algorithm then searches for active subnetworks. The package itself provides provides three algorithms for subnetwork search Greedy, Simulated annealing, and Genetic algorithm. In our case we resorted to Greedy algorithm as it was the default one and recommended by the package authors. In our case the subnetwork search was repeated 10 times. We've updated this part in methods section (lines 146 - 157). Following this, the genes from active subnetworks are then used to perform pathway enrichment analysis; here the initial protein-protein interaction network is also used to derive background genes for the analysis. The enrichment analysis was done using Reactome database. We also updated methods section describing the choice behind focusing on reactome instead of KEGG, wikipathways or other databases. Please see tracked changes in lines 149 - 152. We also added information stating that only the pathways with false discovery rate adjusted p-values lower than 0.05 were considered as significant. Please see tracked changes in lines 154 - 155.

This package also provides the tools to cluster the enriched terms and identify representative terms in each cluster, to score the enriched terms per sample and to visualize analysis results. Regarding the clustering analysis, we performed it using hierarchical clustering method. This was done to cluster the terms according to their biological similarity. We updated this information in methods section, please see tracked changes in lines 155 - 157.

Reviewer 3

Comment 1

1. The first major issue with the study is novelty. Similar studies have been performed earlier. Authors should describe a strong rationale for this study.

Response 1

Thank you for the comment. We understand that there are previous studies that have employed RNA-sequencing to describe gene expression alterations in PanNETs. However, one of the major drawbacks of previous studies is the lack of non-tumor controls. Using this approach we have identified specific pathways that are dysregulated within PanNETs compared to non-tumor pancreatic tissues. One of the most notable findings here is the upregulation of neddylation pathway which has only recently been proposed to serve a significant role in tumor development (PMID: 38575611). Furthermore, the clinical diagnostics of neuroendocrine tumors are in dire need of novel minimally invasive biomarkers. Accordingly, in our data we have identified specific genes that encode secretome proteins and cell surface markers that are upregulated in tumor tissues compared to non-tumor pancreas. These highlighted markers could serve as potential biomarkers for blood-based marker assay development to assist with the diagnostics and prognostics off PanNETs. Therefore we consider that the publishing of this data and observations would only strengthen the research field of pancreatic neuroendocrine tumors.

Comment 2

2. Another issue with the manuscript is lack of details for each figure panel. For example, in Fig. 2, authors write 1 line for all the panels in this figure and mention that a number of genes were differently expressed

---

## [Decision Letter · Decision Letter 1]

Dear Dr. Niedra,

Thank you for submitting your manuscript to PLOS ONE. After careful consideration, we feel that it has merit but does not fully meet PLOS ONE’s publication criteria as it currently stands. Therefore, we invite you to submit a revised version of the manuscript that addresses the points raised during the review process.

We look forward to receiving your revised manuscript.

Kind regards,

Xianmin Zhu

Academic Editor

PLOS ONE

Journal Requirements:

Reviewers' comments:

Reviewer's Responses to Questions

**Comments to the Author**

Reviewer #1: All comments have been addressed

Reviewer #2: All comments have been addressed

Reviewer #3: (No Response)

2. Is the manuscript technically sound, and do the data support the conclusions?

Reviewer #1: Yes

Reviewer #2: Yes

Reviewer #3: Yes

3. Has the statistical analysis been performed appropriately and rigorously?

Reviewer #1: Yes

Reviewer #2: Yes

Reviewer #3: Yes

4. Have the authors made all data underlying the findings in their manuscript fully available?

Reviewer #1: Yes

Reviewer #2: Yes

Reviewer #3: Yes

5. Is the manuscript presented in an intelligible fashion and written in standard English?

Reviewer #1: Yes

Reviewer #2: Yes

Reviewer #3: Yes

Reviewer #1:  It is very good.

Reviewer #2: (No Response)

Reviewer #3: The authors have addressed most of the points. I have few additional points which may require attention. These will help authors improve the reach of manuscript to readers.

1. In Fig. 2A, in the PCA plot, some TAPT and tumor samples look very similar. What could be the reason for this?

2. In Fig. 2B, what value corresponds to which sample?

3. The authors mention neddylation, however NEDD8, the genes that authors mention in this pathway, is not one of the top 25 DEGs.

4. What is neddylation and how is it contributing to the progression of PanNETs? The authors do discuss this briefly in the discussion section, however I believe this requires a more detailed explanation. Does upregulation of neddylation indicate increased proteosome degradation?

5. In the section 'Transcriptomic changes across different grades of PanNETs', the results should be discussed in more details about the DEGs across different tumor types and their relevance.

**Do you want your identity to be public for this peer review?** For information about this choice, including consent withdrawal, please see our Privacy Policy

Reviewer #1: No

Reviewer #2: No

Reviewer #3: No

---

## [Author Response · Author response to Decision Letter 2]

6 May 2025

Responses to comments by Reviewer 3

Comment 1

1. In Fig. 2A, in the PCA plot, some TAPT and tumor samples look very similar. What could be the reason for this?

Response:

Thank you for the comment. I believe this inquiry is related to the two tumor samples that are close to NT tissues. We doubled checked the PCA plot, this time with sample id labels. These two samples are NET22 and NET23 which do not have adjacent non-tumor tissue samples. Therefore, it is rather hard to pinpoint the exact issue why these samples are close to NT samples without additional context from tissue morphology of the tissue section that was used for RNA extraction. Clinically they were both diagnosed G1 tumors which means they retain high level of neuroendocrine differentiation. From technical standpoint it would appear that these two tumor samples which were used for RNA extraction contained significant amount of adjacent non-tumor tissue despite pathologist guided microdissection. We would still like to include these two samples within final results, but we adjusted the results section (please see updated lines 180-182) explaining that there is a possibility that these two tumor samples may contain adjacent non-tumor tissue.

Comment 2

2. In Fig. 2B, what value corresponds to which sample?

Response:

Thank you for the comment. The plot in fig 2B is a p-value histogramm which shows the distribution of P-values from differential expression analysis comparing all PanNET samples vs. all tumor adjacent pancreatic tissues (TAPT) samples, therefore this is comparision not specific sample data. To better explain this we updated the description within fig2 legend. In general, P-value histogramms are one of the ways to check whether there are any problems with multiple hypothesis tests and to get an overall picture of how significant is the difference between two sample groups. In our case the plot shows that p-values are textbook anti-conservative meaning that there is a statistically significant difference in gene expression between PanNETs and TAPT samples and no alarming issues with the DEseq2 analysis parameters we used.

Comments 3 and 4

3. The authors mention neddylation, however NEDD8, the genes that authors mention in this pathway, is not one of the top 25 DEGs.

4. What is neddylation and how is it contributing to the progression of PanNETs? The authors do discuss this briefly in the discussion section, however I believe this requires a more detailed explanation. Does upregulation of neddylation indicate increased proteosome degradation?

Response:

Thank you for comments 3 and 4. NEDD8 was not differentially expressed in our results. We briefly mentioned NEDD8 in discussion section to explain how neddylation system works since the neddylation pathway was the most overrepresented pathway in list of genes differentially expressed in PanNETs. In neddylation, NEDD8 can play a similar role as ubiquitin where NEDD8 ligation alters the stability and function of target protein. NEDD8 also plays an important role in ubiquitination system itself where it is necessery to activate cullin-RING ligases that carry out the ubiquitination process, therefore serving as a molecular switch in the ubiquitination system. When we further explored DEGs related to neddylation using using STRING database we observed that these DEGs form two clusters (S1 Figure). Please see updated sections in lines 239 - 246 in the results section and lines 416 - 423 in the discussion section.

According to Gene Ontology (GO) database Cluster 1 DEGs belong to proteasomal complex an ubiquitin ligase complex GO cellular components and proteolysis involved in protein catabolism GO process. The DEGs forming Cluster 2 on the left represent cullin-RING ubiquitin ligase complex and protein ubiquitination process. This indicates that there is perhaps an upregulation of protein degradation via proteasomes that is regulated by neddylation. High proteasome activity has been observed and discussed in other malignancies and has been suggested as one of the therapy targets (29047025). Interestingly there is little infromation about this in PanNETs, possibly due to the fact that RNA-seq based studies in PanNETs are generally rare. Accordingly, this is one of the first reports where the upregulation of proteasomal activity can also be observed in PanNETs warrating further functional investigation as perhaps targeting proteasomal activity could serve as a novel treatment strategy.

Comment 5

5. In the section 'Transcriptomic changes across different grades of PanNETs', the results should be discussed in more details about the DEGs across different tumor types and their relevance.

Response:

Thank you for the suggestion. Indeed, we should have discussed this more in-depth. Although the changes of gene expression between different grades are less pronounced than tumors versus non-tumors, we did find some interesting observations in G3 vs. G1 tumor comparison that we should have discussed within the initial draft of this manuscript. Here, we can observe that several potential oncogenes (LPC2, MYDGF, PLEKHG2, and FOXQ1) are significantly upregulated in G3 NETs compared to G1 NETs. We updated the discussion section discussing these observations. Please see updated lines 449-467 in document with tracked changes.

---

## [Decision Letter · Decision Letter 2]

Transcriptomic profiling of pancreatic neuroendocrine tumors: dysregulation of WNT, MAPK, PI3K, neddylation pathways and potential non-invasive biomarkers

PONE-D-24-58610R2

Dear Dr. Niedra,

We’re pleased to inform you that your manuscript has been judged scientifically suitable for publication and will be formally accepted for publication once it meets all outstanding technical requirements.

Kind regards,

Xianmin Zhu

Academic Editor

PLOS ONE

Additional Editor Comments (optional):

Reviewers' comments:

Reviewer's Responses to Questions

**Comments to the Author**

Reviewer #3: All comments have been addressed

2. Is the manuscript technically sound, and do the data support the conclusions?

Reviewer #3: Yes

3. Has the statistical analysis been performed appropriately and rigorously?

Reviewer #3: Yes

4. Have the authors made all data underlying the findings in their manuscript fully available?

Reviewer #3: Yes

5. Is the manuscript presented in an intelligible fashion and written in standard English?

Reviewer #3: Yes

Reviewer #3: (No Response)

**Do you want your identity to be public for this peer review?** For information about this choice, including consent withdrawal, please see our Privacy Policy

Reviewer #3: No

---

## [Editor Report · Acceptance letter]

PONE-D-24-58610R2

PLOS ONE

Dear Dr. Niedra,

I'm pleased to inform you that your manuscript has been deemed suitable for publication in PLOS ONE. Congratulations! Your manuscript is now being handed over to our production team.

Kind regards,

on behalf of

Dr. Xianmin Zhu

Academic Editor

PLOS ONE